# Seeking culturally safe care: a qualitative systematic review of the healthcare experiences of women and girls who have undergone female genital mutilation/cutting

Catrin Evans,[1] Ritah Tweheyo,[1] Julie McGarry,[1] Jeanette Eldridge,[2] Juliet Albert,[3] Valentine Nkoyo,[4] Gina Marie Awoko Higginbottom[1]

¹School of Health Sciences, University of Nottingham, Nottingham, UK
²Research and Learning Services, School of Health Sciences, University of Nottingham, Nottingham, UK
³Department of Midwifery, Imperial College Healthcare NHS Trust, London, UK
⁴Mojatu Foundation, Nottingham, UK

**Correspondence to**
Dr Catrin Evans;
catrin.evans@nottingham.ac.uk

## ABSTRACT

**Objective** To explore the experiences of accessing and receiving healthcare related to female genital mutilation/cutting (FGM/C) across the life course from the perspective of women and girls who have undergone FGM/C.

**Design** A systematic review of qualitative research studies using a thematic synthesis approach.

**Methods** Inclusion criteria were qualitative studies (including grey literature) of any design, from Organisation for Economic Co-operation and Development (OECD) countries, of any date and any language. Sixteen electronic databases were searched from inception to December 2017, supplemented by reference list searching. Papers were screened, selected and quality-appraised by two reviewers using established tools from the Joanna Briggs Institute. NVivo software was used to extract study characteristics and code study findings. An inductive thematic synthesis approach was undertaken to identify descriptive themes and interpret these into higher order analytical constructs. Confidence in the review findings was assessed using Grading of Recommendations, Assessment, Development and Evaluations-Confidence in Evidence from Reviews of Qualitative Research (GRADE-CERQual).

**Results** Fifty-seven papers (from 55 distinct studies) from 14 different OECD countries were included (50% published within the last 8 years). One-third of studies focused exclusively on maternity care experiences, whereas others covered a range of foci. No studies reported explicitly on girls' experiences or on experiences of health service-led safeguarding interventions. Only three studies addressed psychological care. The synthesis developed 17 descriptive themes, organised into 5 analytical constructs. These related to communication, access to care, experiences of cultural dissonance/integrity, disempowering care experiences and positive care encounters. The themes illuminate significant challenges to obtaining timely and holistic care (especially for deinfibulation), and highlight different ways in which women may experience care as disrespectful, unsafe and disempowering. Key elements of 'culturally safe care' are identified.

**Conclusions** This review has highlighted key knowledge gaps, especially around (1) girls'/unmarried women's experiences and (2) the impact of recent safeguarding

## Strengths and limitations of this study

► This is the first review that goes beyond a maternity care focus to examine women's/girls' experiences of female genital mutilation/cutting (FGM/C)-related healthcare across the life course, enabling new insights into care-seeking and care access, especially around deinfibulation.

► This is an exceptionally comprehensive review due to its wide focus and its inclusion of grey literature and papers in any language.

► The review has used a theoretical model of 'cultural safety' to inform its analytical approach and has assessed confidence in its findings using Grading of Recommendations, Assessment, Development and Evaluations-Confidence in Evidence from Reviews of Qualitative Research (GRADE-CERQual).

► The review process has been informed by strong community involvement and input from a multidisciplinary expert advisory group at every stage.

► Methodological limitations of the included studies make it difficult to develop findings that reflect nuances of experience related to ethnic group, nationality, age or type of FGM/C.

interventions. There is an ongoing need for community engagement, service development and staff training.

**PROSPERO registration number** CRD420150300012015.

## INTRODUCTION

Health services in higher income countries are said to be operating in a context of 'super-diversity'.[1–5] Whereas in previous decades inmigration was primarily from a limited number of countries traditionally linked through former colonial ties, contemporary migration involves groups from many diverse countries across the globe. Superdiversity poses significant challenges to host country health services, in terms of adapting

services and providing culturally appropriate and accessible care to very many different migrant groups with widely different migration histories, languages, health needs and social vulnerabilities.

Female genital mutilation (or female genital cutting)—hereafter referred to as FGM/C—is practised in 30 countries across North and sub-Saharan Africa and in parts of the Middle East and Asia.[6] In an era of superdiverse migration, European and other high-income countries (eg, USA, Canada, Australia and New Zealand) are becoming home to increasing numbers of women and girls who have experienced, or who may be at risk of, FGM/C (within Europe alone, there are thought to be over half a million women and girls who are FGM/C survivors).[7] These women and girls come from highly diverse regions with different cultural traditions, contexts and beliefs associated with the practice of FGM/C and where different types of FGM/C are practised. For example, in some communities (eg, within Somalia, Somaliland, Sudan, Eritrea and Kenya), type 3 FGM/C is common involving infibulation (narrowing of the vaginal opening through the creation of a covering seal, formed by cutting and repositioning the labia minora, or labia majora, sometimes through stitching, with or without removal of the clitoris). In other communities, however, type 1 (partial or total clitoridectomy) or type 2 (partial or total clitoridectomy and removal of the labia minora, with or without excision of the labia majora) is more common.[8] However, it should be noted that these typologies represent biomedical categories and that different communities have their own nomenclature. Moreover, the exact type of FGM/C a woman has experienced may depend to some extent on the skills and experience of the 'cutter', and women themselves may not always be aware of which type of FGM/C they have experienced.

FGM/C can involve immediate and long-term psychological, sexual, relational and physical health sequelae.[9] Health problems may be particularly severe for women with type 3 FGM/C, who require a degree of deinfibulation in order to have sexual intercourse and to give birth. WHO defines deinfibulation as *'the practice of cutting open the sealed vaginal opening in a woman who has been infibulated, which is often necessary for improving health and well-being as well as to allow intercourse or to facilitate childbirth'.*[8] Currently, deinfibulation is recommended for women and girls reporting medical or psychosexual symptoms related to type 3 FGM/C or on request (ie, personal choice). In addition, global guidelines specifically recommend that deinfibulation is undertaken to prevent obstetric complications, although the optimal timing for the procedure (antepartum or intrapartum) is unclear.[9 9–17] There are compelling clinical reasons however for preferring antepartum deinfibulation, including the fact that it can be performed under local anaesthetic in an outpatient setting, thus reducing costs and risks associated with any emergency procedures that may emerge during labour.[18] In addition, in destination countries where staff may be less familiar with type 3 FGM/C, planned antepartum or premarital deinfibulation ensures that it is performed by trained and experienced professionals.[15]

In destination countries, it is essential, therefore, that care pathways are developed that are able to support women's potential psychological and sexual health needs,[9–11 19 20] as well as supporting decision making around deinfibulation. In addition, health professionals are called on to play a key role in the prevention of FGM/C, through health education with affected communities and identification of girls who may be at risk.[20–22] However, as a cultural phenomenon associated with women's sexuality, gender norms and genital area, FGM/C is a sensitive topic that can be hard for women, communities and health professionals to openly discuss.[23–25] Recent studies of health sector involvement in the management of FGM/C in destination countries show variable availability of specialist services and staff training.[21 26] Moreover, several systematic reviews indicate that health professionals may lack knowledge, confidence and competence in managing FGM/C.[27–30] In addition, there are ongoing calls for greater community involvement in the development of appropriate services related to FGM/C.[31–33]

## Aim

This paper reports the findings of a qualitative systematic review that aimed to explore the experiences of accessing and receiving FGM/C-related healthcare across the life course for women and girls who have undergone FGM/C.[34]

## Rationale

Several prior reviews have been undertaken in relation to this topic; however, we felt a new review was warranted for several reasons. First, most existing reviews have focused on women's maternity care experiences or maternity-focused interventions.[35–41] However, as indicated above, a key issue for services is the development of joined-up care pathways that are able to offer women a range of holistic services at different time points. Hence, our review adopted a focus on the life course. Second, it is well known that migrants experience a range of barriers to accessing health services,[42–46] yet 'access to care' has not been a focus of prior reviews related to FGM/C. Third, several prior reviews have included studies from all over the world, thereby bringing together widely different cultural and health system contexts.[37–39] We felt that a review focused just on 'destination' countries would provide findings that were more transferable to these specific contexts.[47 48] Finally, prior reviews have often limited their searches to English-language papers or to published literature only.[35 36] By contrast, this review has included any language and a wide range of grey literature sources, and thus is able to offer an extremely comprehensive picture.[49 50]

## Theoretical perspective

This review is broadly informed by the theoretical construct of 'cultural safety'.[51] Cultural safety derives from critical social theory and argues that traditional approaches to 'cultural awareness' or 'cultural competence' within healthcare fail to adequately take into account power relations which are historically unequal between migrant or indigenous groups and healthcare providers and services.[52] Within a cultural competence discourse, for example, it is argued that *'the power to define the norm and the onus for action to understand and know about another culture fall to the health professional/service'* (p200).[51] As such, migrant groups remain represented as the 'other' and may be viewed within a deficit model—as a problem that needs to be fixed—rather than as partners whose knowledge and values can contribute equally to a relationship or to service development initiatives. Culturally safe practice, by contrast, is predicated on relationships of mutual trust and respect. It is a transformative and rights-based approach, seeking to uphold the principles of respect, dignity, empowerment, safety and autonomy.[53] Cultural safety envisages the healthcare encounter as a negotiated and equal partnership in which trust plays a central role. The healthcare practitioner's role is to enable patients to say how a service can be adapted and to negotiate an agreed approach—a key aspect of shared decision making.[54] The converse of cultural safety is 'cultural risk'[55] in that culturally unsafe practices are defined as *'any actions that diminish, demean or disempower the cultural identity and well being of an individual or group'* (p7).[56] Cultural safety is not just a feature of individual practice however. It is argued that culturally safe practice must be institutionalised through organisational policies and structures so that it becomes part of mainstream healthcare provision rather than being dependent on individual practitioners who may or may not adopt its values and approach.[57–59] This theoretical perspective is well suited to areas of healthcare involving stigmatised or sensitive topics, such as FGM/C, and where there is a strong need for community engagement in service development and shared decision making in care delivery.

To date, reflexive consideration of the use of theory in qualitative systematic reviews has received minimal attention. Indeed, some authors have referred to the conceptual process of meta-synthesis as a 'black box',[60] and recent research has highlighted the poor reporting of the interpretive work of synthesis.[61] Yet, as noted by Guba and Lincoln,[62] there is no such thing as 'theory-free' research. Some authors report using a 'framework synthesis' approach in which the review is designed to test or expand a pre-existing theoretical framework (which is used as the basis of an initial coding framework to develop key themes).[63] In contrast to this relatively deductive approach, in the present review, the concept of cultural safety was used in three ways to support an interpretive inductive approach. First, it acted as one of a variety of 'lenses' and perspectives (along with other aspects of our identities and backgrounds) through which we interpreted the findings of the research studies.[64] As such, it represented what Maxwell[39 65] has termed an 'idea context' or 'spotlight' for the review process in which a theory can help to identify themes that might otherwise be overlooked. Second, as the review progressed, we felt that concepts related to cultural safety remained a 'good fit' for helping us to understand and position the review findings within the wider literature.[66] Hence, we drew on some of its ideas to inform and structure the discussion and recommendations. Finally, we were aware that a common criticism of systematic reviews is their failure to be policy-relevant or transferable.[67] We felt that framing the discussion and recommendations of the review findings in relation to a theoretical framework might help to enhance the potential transferability of the review findings and guide future research.[66]

## METHODS

This qualitative systematic review is reported following the enhancing transparency in reporting the synthesis of qualitative research (ENTREQ) guidelines.[68] The review was registered in PROSPERO[69] and the methods are documented in a published protocol.[34]

### Search strategy

An exhaustive and sensitive three-step search strategy was designed by an experienced information scientist (JE). First, we searched 11 electronic databases using a combination of index terms and text-based queries. These were searched from inception to a cut-off date of 31 December 2017. An example search strategy for Ovid MEDLINE is provided in online supplementary file 1. Second, we searched for relevant grey literature drawing on five key resources, Google, Google Scholar and suggestions from the project's expert advisory group (see table 1 for all resources/databases that were searched).[70–73] Finally, we hand-searched the reference lists of related systematic reviews and of all the included studies. All retrieved data sets were downloaded into group sets within an EndNote library and duplicates were removed.

### Screening and selection

The review included studies from any date and any language that met the following inclusion criteria: (1) empirical research, (2) qualitative research of any design/methodology (including qualitative findings from mixed methods studies), (3) undertaken in an Organisation for Economic Co-operation and Development (OECD) country (OECD was used as a proxy for potential and comparable high-income 'destination' countries), and (4) needed to explicitly report women's or girl's experiences of seeking, accessing or receiving healthcare associated with FGM/C at any point in the life course and in any clinical setting.

Four team members (RT, CE, JM, GMAH) independently assessed all titles and abstracts against the inclusion criteria. Full-text versions of papers deemed

**Table 1**  List of databases and resources searched

| Electronic databases searched | Date of search |
| --- | --- |
| 1. Ovid multifile search (MEDLINE, EMBASE, PsycINFO) | 10 March 2017 |
| 2. POPline (via http://www.popline.org/), 1970–present | 10 March 2017 |
| 3. ProQuest multifile search | 10 April 2017 |
| 4. Applied Social Sciences Index Abstracts on ProQuest, 1987–current | 26 May 2017 |
| 5. Ovid MEDLINE 1948– and MEDLINE In-Process & Other Non-Indexed Citations to daily update | 26 July 2017 with monthly alert thereafter (cut-off date for included results 31 December 2017) |
| 6. Ovid EMBASE, 1980–2017 week 11 | 3 August 2017 with monthly alert thereafter (cut-off date for included results 31 December 2017) |
| 7. CINAHL Plus with Full Text/EBSCOhost to 2017 | 11 August 2017 with monthly alert thereafter (cut-off date for included results 31 December 2017) |
| 8. Ovid PsycINFO, 1972–March 2017 week 3 | 14 August 2017 with monthly alert thereafter (cut-off date for included results 31 December 2017) |
| 9. MIDIRS on Ovid, 1971–April 2017 | 18 August 2017 |
| 10. HMIC on Ovid, 1979 to date | 18 August 2017 |
| 11. Thomson Reuters Web of Science, 1900–2017 | 18 August 2017 |
| Grey literature sources | |
| 1. British Library Ethos (ethos.bl.uk) | |
| 2. Networked Digital Library of Theses and Dissertations (www.ndltd.org) | |
| 3. National Institute for Health and Care Excellence (https://www.nice.org.uk) | |
| 4. Trove - National Library of Australia (trove.nla.gov.au) | |
| 5. OpenGrey (http://www.opengrey.eu/) | |
| 6. Google | |
| 7. Google Scholar | |
| 8. Experts in the field | |

HMIC, Health Management Information Consortium; MIDIRS, Midwives Information & Resource Service.

to be potentially relevant were obtained and assessed. Papers found not to meet the criteria were excluded with reasons noted (see online supplementary file 2). Any areas of ambiguity were discussed with the wider project team. Non-English-language papers found to be relevant on the basis of their English abstract were sent for complete academic translation.

### Quality assessment

Study quality was assessed by two reviewers using the Joanna Briggs Institute Qualitative Assessment and Review Instrument (JBI-QARI).[74 75] Following the guidance of the Cochrane Qualitative Methods and Implementation Group[76] and the JBI,[49] reports and theses from the grey literature were appraised in the same way as published peer-reviewed papers.[70 72 73] Studies were not excluded on the basis of quality, rather the quality assessment was used to judge the relative contribution of each study to the overall synthesis, and to assess the methodological rigour of each study as part of a process of assessing confidence in the review findings.[77–81]

The JBI-QARI tool was applied to each individual paper and an aggregate score for each was calculated. A criticism of 'scoring' qualitative critical appraisals is that it can be hard to distinguish between the poor conduct of a study and poor reporting, especially where journal word limits constrain the level of detail that can be reported.[82] In addition, there is no consensus regarding the relative importance of any one domain within an assessment tool over another, and hence whether they should all be given an equal weight or not. In view of these concerns, we chose to adopt a 'weighting system' used in previous studies by Higginbottom *et al*,[83 84] in which papers were grouped into one of three 'bands'—high, medium or low—to enable a broad brush evaluation to be made of their relative quality (see table 2).

As an additional strategy for overcoming the potential limitations of solely relying on a checklist to assess quality, we also assessed the 'richness' of the studies. This is an approach outlined by Popay *et al*,[85] and subsequently operationalised in Noyes and Popay[86] and Higginbottom *et al*.[83 84] This approach defines study 'richness' as *'the extent to which the study findings provide explanatory insights that are transferable to other settings'* (p230).[86] 'Thick' papers create or draw on theory to provide indepth explanatory insights that can potentially be transferable to other contexts. By contrast, 'thin' papers provide limited or

**Table 2** Quality evaluation bands

| Quality evaluation | JBI-QARI tool aggregate score | Definition |
|---|---|---|
| High | Over 7 | A study with a rigorous and robust scientific approach which meets most JBI benchmarks (*perhaps 7 or more 'Yes'*). |
| Medium | Between 5 and 7 | A study with some flaws but not seriously undermining the quality and scientific value of the research conducted (*perhaps 5–7 'Yes'*). |
| Low | Under 5 | A study with flaws and poor scientific value (*perhaps below 5 of the benchmarks met*). |

JBI-QARI, Joanna Briggs Institute Qualitative Assessment and Review Instrument.

superficial description and offer little opportunity for generalising. Each paper was assessed against the criteria as set out in Higginbottom *et al* (p5)[83] (see table 3) and categorised as either 'thick' or 'thin'.

### Data extraction and assessment of relevance

Study characteristics were extracted by one author (RT) using a modified JBI template and double-checked by CE. During this process, papers were categorised in terms of their relevance to the review question. This assessment was made in order to gain a better understanding of the nature of the body of evidence, and also to facilitate the coding process, as described further below. Study relevance was defined as high, medium or low, as set out in table 4. PDFs of all the included papers were imported into NVivo software, and the 'findings/results' and 'discussion' sections were coded and analysed.[87]

### Data analysis and synthesis

The review adopted a thematic synthesis approach as outlined by Thomas and Harden[88] involving four iterative stages: (1) indepth reading of the whole papers, (2) inductive line-by-line coding of the findings, (3) grouping the codes on the basis of shared characteristics and meanings into descriptive themes, and (4) interpreting higher order analytical themes.

Studies categorised as 'thick', high-quality and 'highly relevant' were used as index papers to develop an initial set of 'open codes'.[89–101] Two reviewers (RT, CE) undertook this task independently. The codes were then reviewed and compared. Where reviewers had identified and coded the same issue, a code for that issue was agreed for use in coding subsequent papers. A standardised name was identified for the code and a description was

produced. Where reviewers had applied slightly different codes to a concept indicating a different interpretation of meaning, they discussed it and either agreed a shared code (based on a shared understanding of meaning) or created two different codes to reflect two different meanings. The initial set of codes were 'free nodes'. These were compared, analysed and discussed by the whole team. Where meanings appeared to relate to a similar concept, these were grouped into broad descriptive themes and subthemes. This process created a codebook which was applied to the remaining papers and expanded, refined and modified as appropriate (by identifying new codes and new themes, or by merging and renaming existing codes and themes). The codebook ensured that definitions of codes and themes were explicit and could be easily shared across the team and discussed. More details on this process are given in online supplementary file 3, using the analytical theme of 'Communication is Key' (see below) as an exemplar. Analytical themes were evolved through an indepth process of comparing and contrasting the meanings of the descriptive themes, analysing these in relation to how they were, or were not, able to illuminate the review questions, and inferring broader phenomena, categories of meaning or social processes that they related to.[102 103]

### Patient and public involvement

Public involvement was an integral part of the review process and was achieved in three ways. First, the review was conceived and co-constructed from an ongoing partnership between an academic team (CE, RT, GMAH, JM, JE), clinical experts (JA) and a community organisation working on FGM/C-related issues (VN). By undertaking

**Table 3** Assessment of study richness

| Richness | Operational definition |
|---|---|
| Thick papers | ► Offer greater explanatory insights into the outcome of interest.<br>► Provide a clear account of the process by which the findings were produced—including the sample, its selection and its size, with any limitations or bias noted—along with clear methods of analysis.<br>► Present a developed and plausible interpretation of the analysis based on the data presented. |
| Thin papers | ► Offer only limited insights.<br>► Lack a clear account of the process by which the findings were produced.<br>► Present an underdeveloped and weak interpretation of the analysis based on the data presented. |

**Table 4** Assessment of study relevance

| Study relevance | Definition |
| --- | --- |
| High (specific) | FGM/C-specific healthcare (eg, the study is focused on a specific aspect of care related directly to FGM/C, eg, deinfibulation, childbirth for women who have had FGM/C, psychological care). |
| Medium (direct) | Other healthcare context (eg, where the study focus is on the maternity care experience of a particular group more generally and where some of the findings relate to the experience of FGM/C). |
| Low (indirect) | Where the study focus is on general attitudes towards FGM/C and/or experiences and consequences of FGM—and where some FGM/C-related healthcare issues are reported, but are not the main focus of the paper. |

FGM/C, female genital mutilation/cutting.

the review we hoped to illuminate women's voices and experiences in order to inform the provision of FGM/C-related healthcare. Second, an expert advisory group was established from the outset, which included FGM/C survivors, activists and health professionals. This group helped with identifying relevant literature and participated in project meetings in which the synthesis and recommendations were formulated. Finally, the review findings were shared with a wide range of FGM/C activist and survivor groups and community organisations, and their feedback helped to shape the final report and recommendations.

### Rigour within the analytical process

To ensure rigour of the analytical process, the team sought to identify and understand possible 'disconfirming' cases that might challenge emerging interpretations,[104] and to explore possible subgroup or contextual differences. These processes were aided by the creation of a theme matrix (see online supplementary file 4), in which each theme was mapped to its constituent studies. This helped the team to see clearly how common the theme was among the studies, what kind of study contexts or samples the theme related to, and to explore why it may have been present in some studies but not in others. In addition, as described above, the whole team and wider project advisory group contributed to the evolution of the synthesis, through reading of key papers and providing feedback on the emerging interpretations.

### Assessment of confidence in the review findings (CERQual)

Assessment of confidence in the findings of the review was undertaken using the Confidence in Evidence from Reviews of Qualitative Research (CERQual) approach.[76 77 105–111] Akin to Grading of Recommendations, Assessment, Development and Evaluations (GRADE), CERQual uses assessments of 'concerns' within four distinct domains (methodological limitations, relevance, coherence and adequacy of data) applied to each individual review finding. The CERQual assessment was applied to the descriptive rather than analytical themes, as the latter comprise an aggregation of descriptive themes as well as descriptions of patterns or explanations that are inferred as part of the interpretative process. There is, as yet, relatively little guidance for,

or experience with, applying CERQual to higher level analytical themes or theories.[108]

## RESULTS

### Search outcome

The search outcomes are reported in detail in the Preferred Reporting Items for Systematic Reviews and Meta-Analyses flow chart in figure 1. Fifty-seven papers met the inclusion criteria, representing 55 distinct studies. Fifty-three papers were in the English language, two were published in German[112 113] and two in Spanish.[114 115]

### Study characteristics

Due to the large numbers of studies in this review, a highly summarised description of key study characteristics and key methodological assessments is found in table 5. Full details are provided in online supplementary files 5 and 6.

The studies represented 14 different countries, including Australia,[100 101 116 117] Austria,[113] Canada,[118–123] Finland,[124] France,[90] Germany,[112 125] the Netherlands,[126 127] New Zealand,[116 128] Norway,[99 129] Spain,[114 115] Sweden,[23 94 130–132] Switzerland,[98 133] Scotland,[89 134 135] England[25 92 93 95–97 136–144] and USA.[91 145–153] The studies were a mixture of older and more recent research, with publication dates ranging from 1985 to 2017. However, 29 papers had been published since 2011; hence, half of the papers reflected a more contemporary context. Over half of the included papers (n=33) were peer-reviewed journal articles.[23 25 90 92 94 95 97 99 114 116 117 119 122 124 126 127 129–133 136 137 139 144–147 149–153] Other papers included 13 theses,[89 91 93 98 112 113 115 118 120 121 123 128 148] 10 unpublished research reports[96 100 101 125 134 135 140–143] and 1 book chapter.[138]

A large proportion of studies (n=18) focused specifically on women's birth/maternity care experiences.[89–91 94 95 97 99 117 119 128 130–132 139 145 146 150] Other studies focused on general views on healthcare,[25 100 101 112 113 115 122 138 140 152 153] general attitudes towards FGM/C,[23 96 121 123 125 134–136 142 143 151] experiences of sexual/reproductive health services,[98 114 116 124 133 148] cervical screening,[126 137 147 149] psychological issues,[92 93 127] deinfibulation,[129 144] general practitioner (GP) services,[141] identity[118] and pain/embodiment.[120] There were no studies that examined women's

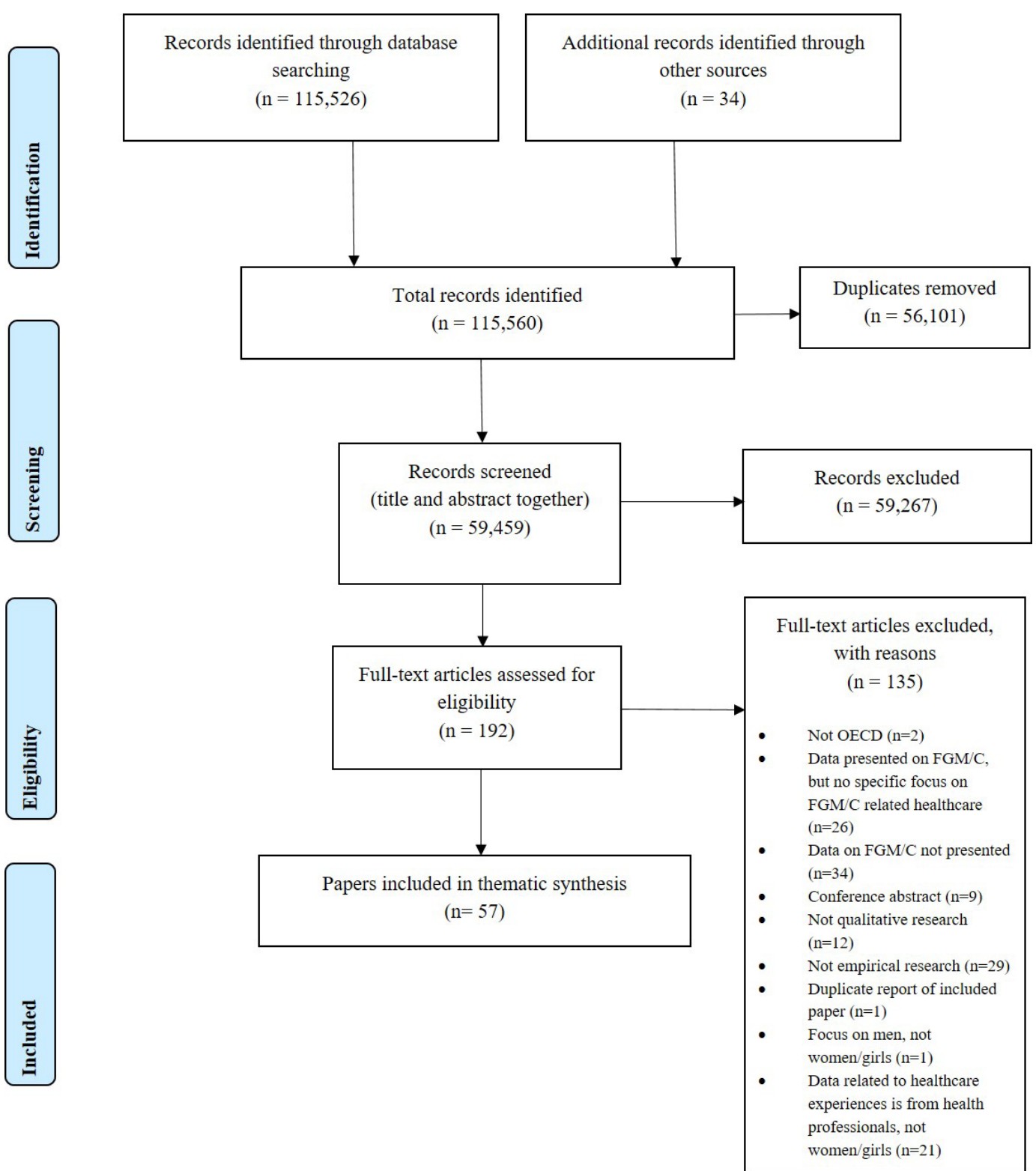

**Figure 1** PRISMA flow diagram. FGM/C, female genital mutilation/cutting; OECD, Organisation for Economic Co-operation and Development; PRISMA, Preferred Reporting Items for Systematic Reviews and Meta-Analyses.

experiences of surgical reconstruction following FGM/C and no studies that examined experiences of health service-led safeguarding interventions.

The studies were primarily of adult women. Only one study included girls under the age of 18 years but did not report health-related experiences of this group.[23] There were no studies that focused specifically on healthcare issues related to FGM/C in older (postchildbearing) women. Hence, although the aim of the review was to explore FGM/C across the life course, the studies all focused on generic issues relating to adult women and no further age-specific differentiation could be made. The majority of studies were

**Table 5** Summary study characteristics and methodological assessments

| Number | Authors (year) | Country | Study focus | Eligible participants and country of origin | Quality assessment | Richness | Relevance |
|---|---|---|---|---|---|---|---|
| 1 | Abdi (2012)[136] | UK | FGM/C and identity for Somali women. | 16 Somali women | High | Thin | Low |
| 2 | Abdullahi et al (2009)[137] | UK | Cervical screening. | 50 Somali women | Medium | Thin | Medium |
| 3 | Ahlberg et al (2004)[23] | Sweden | FGM/C tradition in destination countries. | 50 Somali women | High | Thin | Low |
| 4 | Ahmed (2005)[138] | UK | Cross-cultural psychology and women's experience of FGM/C. | 56 women (from Somalia and Eritrea) | Low | Thin | Low |
| 5 | Ameresekere et al (2011)[145] | USA | Maternity care experiences. | 23 Somali women | Medium | Thin | Medium |
| 6 | Asefaw (2007)[112] | Eritrea and Germany | Postmigration attitudes towards FGM/C. | 31 Eritrean women | Medium | Thin | Low |
| 7 | Baldeh (2013)[89] | Scotland | Maternity care experiences. | 7 women (from Gambia, Ghana and Sudan) | High | Thick | High |
| 8 | Ballesteros et al (2014)[114] | Spain | Health impacts of FGM/C. | 9 Senegalese women | Medium | Thin | Medium |
| 9 | Behrendt (2011)[125] | Germany | Views on FGM/C. | 91 women (from Ghana, Nigeria and Togo) | Medium | Thin | Low |
| 10 | Beine et al (1995)[146] | USA | Maternity care experiences. | 14 Somali women | Medium | Thin | Medium |
| 11 | Berggren et al (2006)[130] | Sweden | Maternity care experiences. | 21 women (from Eritrea, Somalia and Sudan) | Medium | Thick | High |
| 12 | Betts (2011)[118] | Canada | Identity, gender, sexuality and healthcare experiences. | 13 Somali women | Medium | Thick | High |
| 13 | Bravo Pastor del Mar (2014)[115] | Spain | Views on FGM/C. | 24 women (from Senegal, Nigeria and Gambia) | High | Thin | Low |
| 14 | Bulman and McCourt (2002)[139] | UK | Maternity care experiences. | 12 Somali women | Medium | Thin | Medium |
| 15 | Carroll et al (2007)[147] | USA | Healthcare experiences. | 34 Somali women | High | Thick | Medium |
| 16 | Chalmers and Omer-Hashi (2002)[119] | Canada | Maternity care experiences. | 415 Somali women | Medium | Thick | Medium |
| 17 | Degni et al (2014)[124] | Finland | Reproductive health and maternity care experiences. | 70 Somali women | High | Thin | Medium |
| 18 | d'Entremont et al (2014)[90] | France | Childbirth experiences. | 4 women (from different countries) | High | Thick | High |
| 19 | Essén et al (2000)[131] | Sweden | Maternity care experiences. | 15 Somali women | Medium | Thin | Low |
| 20 | Fawcett (2014)[91] | USA | Views on FGM/C and maternity care. | 30 Somali women | High | Thick | High |
| 21 | Gali (1997)[148] | USA | Reproductive healthcare experiences. | 26 women (from Sudan, Eritrea, Ethiopia and Egypt) | Medium | Thin | Low |
| 22 | Ghebre et al (2015)[149] | USA | Views on cervical screening. | 23 Somali women | High | Thick | Medium |
| 23 | Glazer (2012)[120] | Canada | Pain and embodiment associated with FGM/C. | 14 Somali women | High | Thin | Low |
| 24 | Glover et al (2017)[92] | UK | Psychological impact, resilience and experiences of FGM/C. | 20 women (from Somalia, Kenya, South Africa and Gambia) | High | Thick | High |
| 25 | Guerin et al (2006)[116] | Australia and New Zealand | Reproductive healthcare experiences. | 10 women (from Somalia, Ethiopia, Eritrea, Sudan and Nigeria) | Low | Thin | Low |
| 26 | Hill et al (2012)[150] | USA | Maternity care experiences. | 18 Somali women | Medium | Thick | Low |
| 27 | Hussein (2010)[140] | UK | Views on FGM/C and related services. | 8 women (from Somalia and Sudan) | Medium | Thin | Low |

Continued

**Table 5** Continued

| Number | Authors (year) | Country | Study focus | Eligible participants and country of origin | Quality assessment | Richness | Relevance |
|---|---|---|---|---|---|---|---|
| 28 | Hussen (2014)[128] | New Zealand | Views on FGM/C and healthcare experiences. | 20 women (from Eritrea, Ethiopia, Somalia and Sudan) | Medium | Thick | Medium |
| 29 | Johansen (2017)[129] | Norway | Experiences and perceptions of premarital deinfibulation. | 23 women (from Somalia and Sudan) | High | Thick | Medium |
| 30 | Jones (2010)[93] | UK | Views and experiences of FGM/C. | 6 women (from Sudan and Somalia) | High | Thick | High |
| 31 | Khaja (2004)[121] | Canada and USA | Views and experiences of FGM/C. | 17 Somali women | High | Thin | Low |
| 32 | Khaja et al (2010)[122] | Canada and USA | Views and experiences of FGM/C. | 17 Somali women | Low | Thin | Low |
| 33 | Lundberg and Gerezgiher (2008)[94] | Sweden | Maternity care experiences. | 15 Eritrean women | High | Thick | High |
| 34 | Maier (2003)[113] | Austria | Views, experiences and impacts of FGM/C. | 3 women (from different countries) | Low | Thin | Low |
| 35 | Maternity Action (2014)[141] | UK | Maternity care experiences. | 23 women (from several countries) | Low | Thin | Medium |
| 36 | McNeely and Christie-de Jong (2016)[151] | USA | Views and experiences of FGM/C. | 13 Somali women | High | Thin | Low |
| 37 | Moxey and Jones (2016)[95] | UK | Maternity care experiences. | 10 Somali women | High | Thick | High |
| 38 | Murray et al (2010)[117] | Australia | Birth experiences. | 10 women (from Sudan Liberia, Ethiopia and Somalia) | High | Thick | Medium |
| 39 | Norman et al (2016)[142] | UK | Views and experiences of FGM/C and community engagement. | 33 women (from various countries) | Medium | Thin | Low |
| 40 | Norman et al (2009)[143] | UK | Views and experiences of FGM/C, community engagement and services. | 40 women (from Sudan, Eritrea and Somalia) | Medium | Thin | Low |
| 41 | O'Brien et al (2017)[134] | Scotland | Views and experiences of FGM/C, community engagement and services. | 48 women (from Sudan, Nigeria and Zimbabwe) | High | Thin | Low |
| 42 | O'Brien et al (2016)[135] | Scotland | Views and experiences of FGM/C, community engagement and services. | 39 women (from 17 African countries) | High | Thin | Low |
| 43 | Palfreyman et al (2011)[96] | UK | Views and experiences of FGM/C, community engagement and services. | 105 women (from Somali, Eritrea, Sudan, Sierra Leone and Gambia) | High | Thick | High |
| 44 | Recchia and McGarry (2017)[25] | UK | Views on FGM/C and healthcare experiences. | 6 women (from Kenya, Nigeria and Ethiopia) | Medium | Thick | Low |
| 45 | Safari (2013)[144] | UK | Women's experience of deinfibulation. | 9 women (from Somalia and Eritrea) | High | Thin | Low |
| 46 | Salad et al (2015)[126] | The Netherlands | Cervical screening. | 46 Somali women | High | Thick | Medium |
| 47 | Shaw (1985)[152] | USA | Views on FGM/C and healthcare experiences. | 12 women (from Sudan, Egypt and Somalia) | Low | Thin | Low |
| 48 | Shermarke (1996)[123] | Canada | Views on FGM/C. | 8 Somali women | High | Thin | Medium |
| 49 | Straus et al (2009)[97] | UK | Maternity care experiences. | 8 Somali women | High | Thick | High |
| 50 | Thierfelder (2003)[98] | Switzerland | Gynaecological and maternity care experiences. | 29 women (from Somalia and Eritrea) | High | Thick | High |
| 51 | Thierfelder et al (2005)[133] | Switzerland | Gynaecological and maternity care experiences. | 29 women (from Somalia and Eritrea) | Medium | Thin | Medium |

**Table 5** Continued

| Number | Authors (year) | Country | Study focus | Eligible participants and country of origin | Quality assessment | Richness | Relevance |
|---|---|---|---|---|---|---|---|
| 52 | Upvall et al (2009)[153] | USA | Health experiences. | 23 Somali women | Medium | Thick | High |
| 53 | Vangen et al (2004)[99] | Norway | Maternity care experiences. | 23 Somali women | High | Thick | High |
| 54 | Vaughan et al (2014)[100] | Australia | Views and experiences of FGM/C, community engagement and services. | 87 women (from Eritrea, Ethiopia, Somali and Sudan) | High | Thick | High |
| 55 | Vaughan et al (2014)[101] | Australia | Views and experiences of FGM/C, community engagement and services. | 50 women (from Togo, Sudan and Kenya) | High | Thick | High |
| 56 | Vloeberghs et al (2012)[127] | The Netherlands | Psychosocial and health experiences. | 66 women (from Somalia, Sudan, Eritrea, Ethiopia and Sierra Leone) | High | Thin | Medium |
| 57 | Wiklund et al (2000)[132] | Sweden | Maternity care experiences. | 9 Somali women | Medium | Thin | Medium |

FGM/C, female genital mutilation/cutting.

with women from FGM/C practising countries in sub-Saharan Africa, and the majority of these included women specifically from countries in the Horn of Africa, where type 3 FGM/C is most commonly practised. It is likely therefore that the findings included in the review reflect issues that may be specific to these population groups and to the experience of type 3 FGM/C. Only three studies explicitly reported including women from Egypt/Middle East but did not differentiate these women's experience from the rest of the sample.[138 148 152]

### Methodological quality

Full details of assessments of methodological quality, richness (thick/thin) and relevance are provided in online supplementary file 6 (see table 5 for a summary). Thirty papers were assessed as being of high quality,[23 89–101 115 117 120 121 123 124 126 127 129 134–136 144 147 149 151] 21 as medium quality[25 112 114 118 119 125 128 130–133 137 139 140 142 143 145 146 148 150 153] and 6 as low quality.[113 116 122 138 141 152]

A methodological weakness across many studies was lack of discussion of the underpinning philosophical standpoint (question 1 of the QARI tool), making it difficult to assess the congruency of the chosen methodology. Likewise, many studies did not describe any clear methodology (simply stating that they adopted a generic 'qualitative approach'). Such studies tended to have 'thin' or mainly descriptive findings, and it was difficult to judge the congruence of the methodology with the research question and the methods (questions 2 and 3 of the QARI tool). Finally, a limitation across many studies was lack of discussion of reflexivity (questions 6 and 7 of the QARI tool). Given the sensitive nature of FGM/C (and sexuality or migrant healthcare) as a topic, the failure to explore the researcher's own theoretical standpoint or their role, professional background, ethnicity, experience of FGM/C or relationship to the participants is a significant weakness, making it hard to judge the dependability of the findings.[154]

### Thematic synthesis findings

The findings from 57 papers were synthesised into five analytical themes which represent a synthesis and interpretative analysis of 17 descriptive themes. Table 6 shows their inter-relationship and provides one or two quotes to illustrate each descriptive theme. Hence, the description of themes below does not include quotes. Rather, each analytical theme is explained, followed by an elaboration of its constituent descriptive themes. Due to the large number of studies that contributed findings to each descriptive theme, rather than 'crowd' the text with multiple repetitive references to these individual studies, the reader is referred to a theme matrix (see online supplementary file 4) that shows which studies have contributed findings to each theme.

### Analytical theme 1: communication is key

The synthesis revealed that communication was a key interpersonal process that underpinned women's ability

**Table 6** Themes, quotations and CERQual assessment

| Theme number | Theme heading | Studies (n) | CERQual assessment | Indicative quotes |
|---|---|---|---|---|
| | **Analytical theme 1: communication is key** | | | |
| 1.1 | Language barriers and interpretation challenges | n=31<br>23 90 91 95–97 99–101 117–119 124 126–128 130 132–135 137 139–141 143 147–150 153 | High confidence | "I wish I had a way to communicate with the doctor directly. There are things that sometimes I want to say or ask but I feel embarrassed saying it through a translator, especially on the phone…They never translate what I'm saying. They tell me that is a shameful question…" (Sweden, p364)[153] |
| 1.2 | 'Can't talk, not asked': double silence and cultural taboo | n=37<br>25 89–91 93–96 99–101 112 115 117 118 120 121 123–128 130 132–135 138 140–146 151 153 | High confidence | "If they had asked me I would have told them, but I was never asked…I was worried and scared during my first pregnancy because I have heard people talk about the pain but I didn't know how to tell them…I was shy." (Scotland, p23–24)[89] |
| 1.3 | Cultural (in)sensitivity | n=34<br>25 89 91 92 96–101 112 115–122 125 127 128 130 132 135 136 138–143 146–148 153 | Moderate confidence | "The doctor was not comfortable with me. He asked if I enjoy violent sex with my husband. I don't feel comfortable to talk about sex even with females— yet here is a male doctor examining my genitals and making mean comments." (USA, p274)[119] |
| | **Analytical theme 2: access to care: influenced by an interaction of multilevel community and health service processes** | | | |
| 2.1 | Influence of cultural norms | n=35<br>93–95 97 99–101 112 113 117–121 124 126–130 132 133 137 139–141 143–147 149 150 153 | Moderate confidence | "Even though we had FGM type 2 or 3, had a small vaginal opening and were virgins, we did not ask for deinfibulations before marriage because that is shameful in our culture. Most of us who have FGM had problems with penetration following marriage." (New Zealand, p43)[128]<br><br>"In my country normally…people don't go…usually people go to a psychologist if it is something like erm… maybe chronic issues that they cannot solve at all…or sometimes if it is abnormal…They think if you go to a psychologist you are mad or crazy." (England, p6)[93] |
| 2.2 | Influence of the family | n=19<br>90 91 93 94 97 99 100 113 118 121 124 126 130 132 136 144 149 150 153 | Moderate confidence | "I know my friend she wants to do that (deinfibulation) but her husband doesn't let her…I told her, this is good for you - go and do. She said ok but her husband said no…Maybe if she did this (deinfibulation), it would have been a big problem in the family." (England, p157)[144] |
| 2.3 | Knowledge and information about FGM/C services | n=34<br>92 93 95–97 99–101 113–119 121 126 128 130–135 137 139 142 143 145 148–150 152 153 | Moderate confidence | "They need to have someone that talks to you before and while you have a baby. I would have wanted someone to help me and tell me what was going to happen but I never got that…You know someone who was specially trained so you only see people who understand FGM." (England, p233)[92] |
| 2.4 | 'Hit and miss' care | n=38<br>88 91 92 94–97 99–101 112 113 115 117–120 123–125 127 128 130–135 139–143 147 148 150–152 | High confidence | "Sometimes people go to one doctor after another describing their problems to them. They do not understand so they cannot find a solution. There was a woman who was suffering from bleeding and pain. When she went to the GP, he told her it was normal. He did not know how to treat her. She kept going several times and at last they opened her." (England, p41)[143] |
| | **Analytical theme 3: cultural and bodily dissonance: striving for cultural and bodily integrity** | | | |
| 3.1 | Moving from normal to different | n=40<br>89 91–94 96 97 99–101 112 113 115–120 122–130 132–135 140 141 143 144 148 150–153 | High confidence | "Okay, now I'm missing something that other women has…Well the first shock was to find out that people here are not circumcised…You have to deal with that…I'm not the same as women here…and then I started to kind of – you grieve a bit, that 'what is this thing that has been taken away from me?'…For me it has caused me lots of problems because when I went to health services, the minute that I say 'I was circumcised' it just create huge barrier between them and me, like you know - suddenly I'm from another planet." (Australia, p23)[100] |
| 3.2 | Threat to the self: reliving FGM/C pain during clinical interventions | n=28<br>89–92 94 95 99–101 112–114 119 120 126–128 130 133 134 136 137 140 143 148 149 151 152 | Moderate confidence | "You always remember it. They decide to cut me, and when she went to do it she didn't explain to me and I panic. I nearly kick her. Two of the nurses held me down. That was it…I was a little girl again being held down." (England, p228)[92] |
| 3.3 | Being opened: complexities around deinfibulation | n=28<br>89 92–97 100 101 112 113 115 116 118 120 121 124 128–130 133 134 143 144 146 148 150 152 | Moderate confidence | "She [Somali lady] hadn't talked to friends about it, just the midwife…She said: 'When I met the midwife she told me a lot about FGM. The midwife said to me 'If you want I can open it for you. If you don't want that I won't'. We talked about it a lot. After fourmonths I tell her 'I want you to open it for me. Do me a small operation." (Scotland, p31)[134] |

Continued

**Table 6** Continued

| Theme number | Theme heading | Studies (n) | CERQual assessment | Indicative quotes |
|---|---|---|---|---|
| 3.4 | Being changed: complexities around reinfibulation | n=19<br>92 94 96 98 100 101 112 113 115 116<br>118 124 129 130 134 144 146 148 152 | Moderate confidence | "I became a victim in Sudan already when I was 4 years old; I had no choice. Now I have to become a victim again after delivery, when the midwives refuse to resuture me. I just ask for a few stitches, not to have an open wound." (Sweden, p53)[130] |
| **Analytical theme 4: disempowering care encounters** | | | | |
| 4.1 | Being exposed and humiliated | n=40<br>25 89–92 95–97 99–101 112 113<br>115–120 122 125–128 130 132–138<br>140–143 146 148 151 153 | High confidence | "Me – I was installed with my legs spread apart - they all came to see…and I didn't see any reason why they came to see me except for the fact that I was excised. All those people who came, who entered into my intimacy without my authorization, who didn't ask me if it bothered me…I felt that it was an aggression." (France, p308)[90]<br>"They showed no respect to me, I understand now why people don't want to talk about it to the midwife. I will tell anyone not to tell them about being circumcised." (Scotland, p23)[89] |
| 4.2 | Being judged and stereotyped | n=36<br>23 25 89–93 96 97 99–101 112 113<br>115–119 122–125 127 130 132 133<br>139–143 145 147 148 153 | High confidence | "One participant…described that a German gynaecologist (a woman) looked at her with so much pity and irritation after the physical examination that she felt for the first time in her life that she was not normal…she felt that she was treated like a victim who could not be talked to rationally. The participant decided…to abstain from consulting German doctors." (Germany, p83)[125] |
| 4.3 | Lacking choice, power and control | n=32<br>89–92 94 95 97 99–101 112 113 116–121<br>123 125 127 130–134 139 141 143 145<br>146 148 150 | High confidence | "I can't say that the birth went well. The midwife said that I couldn't give birth without being cut. otherwise the baby won't come without a caesarean section or I can lose the baby. Nobody asked me what I thought. I was there in pain…without pain relief, nothing." (France, p308)[90] |
| 4.4 | Feeling unsafe and vulnerable | n=46<br>23 25 89–97 99–101 112–121 123 125<br>126 128 130–135 139–141 143 145–148<br>150–153 | High confidence | "They didn't know what they were doing, and uh, they, you know they had to give me more cuts then in an awkward way…I was telling the doctor to cut the FGM and totally he ignored me. And he was cutting in the sides instead of cutting the front. It was very degrading and I am very distressed to think about it." (Australia, p25)[100] |
| **Analytical theme 5: positive care encounters** | | | | |
| 5.1 | Trusting and appreciating providers and the system | n=34<br>90 92–96 99–101 112–115 117–120<br>123–125 127 128 130–132 134 135 139<br>143 145 146 148 150 153 | High confidence | "I was lucky when I met a midwife in Sweden who knew about circumcised women. This was a great help to make me feel secure because it was my first time to be pregnant and to live far from my parents and family." (Sweden, p219)[94]<br>"I know a friend that had to be opened before her delivery. She was lucky because she had found a midwife who knew about FGM - and she booked her to have the operation before the delivery." (England, p15)[140] |
| 5.2 | Voicing healthcare needs and preferences | n=45<br>25 89–97 100 101 112 113 115 116<br>118–120 122–124 127–130 132–135<br>137–143 145–148 150–153 | High confidence | "A while ago, all the women used to choose caesarean to give birth but now the health services know about women who are circumcised. They know of their situations and the complications they can have during maternity like bleeding and that they need help with the opening before the baby comes. When the health services don't know about your condition, they refer people to other services. There are some who are understanding, especially if different communities come to attend the clinics or hospitals, as they get more oriented on the different cultures of the communities." (England, p43)[143] |

CERQual, Confidence in Evidence from Reviews of Qualitative Research; FGM/C, female genital mutilation/cutting.

to seek care, to obtain care and to have a positive care experience. Conversely, difficulties in communication negatively impacted all of these dimensions. Communication about FGM/C was characterised by issues that might be expected within migrant women's encounters with new host country healthcare systems—such as language barriers within consultations. However, communication about FGM/C presented an additional layer of complexity related to the fact that FGM/C was seen by both women and healthcare providers as a deeply personal, private and sensitive issue touching on a range of taboos associated with gender, culture and sexuality—all issues that in many societies are shrouded in secrecy and silence. When women encountered providers who were able to overcome taboos and cultural differences, it forged bonds of trust and facilitated an environment in which women were able to talk about their FGM/C and to explore options for care. However, the majority of studies reported challenges and problems with communication and interpretation which led to women being unable to talk about their problems or to explore appropriate treatment options, led them to avoid healthcare in general and generated a sense of mistrust in the system. This theme mainly related to maternity care interactions, but also to communication about cervical cancer screening.[100 101 126 134 141]

### Theme 1.1: language barriers and interpretation challenges
The majority of studies (n=31) reported that for women who did not speak the host country language well, communication about FGM/C was hindered by language barriers and problems with accessing interpretation support that was appropriate for discussing highly sensitive and personal issues. Language barriers meant that women were unable to form a trusting relationship with their provider, express their needs adequately or understand information or advice. This led to frustration and increased anxiety, especially in a context where consultation times were often limited.[128 139] In some cases, women felt that they had had poor clinical experiences as a direct result of their inability to communicate about FGM/C.[23 128 134] In other examples, women avoided mentioning their FGM/C at all because they knew they lacked the language to explain.[128 133 140]

### Theme 1.2: 'Can't talk, not asked': double silence and cultural taboo
As a private, sensitive and taboo issue, communication around FGM/C was hindered by a double silence. Women reported that FGM/C was rarely discussed within their own communities, and likewise they found it hard to discuss it with health providers, especially if these (or the interpreters) were male. For this reason, women preferred the topic to be raised by health providers; however, a commonly reported finding was that, even when women appropriately accessed relevant services (eg, GPs or antenatal services[95 142 143]) and might appreciate the general care offered,[115 124] healthcare providers often did not ask about FGM/C, even when the consultations were for

pregnancy-related check-ups. The consequence was that FGM/C was sometimes not identified or discussed until women presented in the labour room, so opportunities for care planning or birth planning were missed.[89] In addition, some studies reported that women found communication to be easier when supported by a knowledgeable community advocate or confident peer.[89 96 118 141 147] Other studies emphasised that open communication was predicated on being able to develop a trusting relationship with the health professional, which in turn required time and continuity of care.[95 100 101 139]

### Theme 1.3: cultural (in)sensitivity
This theme was reported by 34 studies in which women described experiencing comments and questions from health providers that were perceived as clumsy, insensitive or intrusive. As a result, women reported feeling ashamed, scared or stigmatised. Such encounters inhibited women from talking about their FGM/C, and in some cases led to them avoiding examinations or healthcare services.[112 148] By contrast, culturally sensitive communication was greatly appreciated and enabled women to openly discuss their FGM/C and related questions.[90 95 135]

### Analytical theme 2: access to care: influenced by an interaction of multilevel community and health service processes
The review found that access to care related to FGM/C was influenced by an interaction of factors operating at individual and community levels, as well as at health system and service levels. At individual and community levels, studies showed that care-seeking was influenced by cultural norms around health and sexuality, collective approaches to decision making, and the level of knowledge and information that individuals and communities had about FGM/C and health service availability. These factors influenced whether or not a woman would seek care, at what time period and where. However, the studies also showed that care-seeking choices were shaped by the types of services available and that outcomes were influenced by the extent to which the health system, health services and health practitioners were ready and able to provide the range of support that women may need at different points in their lives. The review showed that there were challenges within communities in terms of seeking care and finding the right services. However, even when services were appropriately accessed, care was sometimes haphazard and suboptimal due to variable levels of staff expertise and inconsistent and unclear referral pathways, policies and procedures.

### Theme 2.1: influence of cultural norms
Women's care-seeking in relation to FGM/C was strongly influenced by wider cultural norms around sexuality and health, including cultural norms on the importance of premarital virginity, avoiding male health providers and lack of familiarity with preventive care-seeking. These factors often led women to avoid seeking care unless

symptomatic or pregnant. Thirty-five studies contributed to this theme.

Women reported that strong cultural imperatives valuing premarital chastity and virginity meant that they would avoid procedures requiring gynaecological examinations (eg, Pap smears[101] [126] [143] [149]), and especially for those with type 3 FGM/C the general norm was to avoid seeking deinfibulation prior to marriage and pregnancy due to the cultural pressure to maintain chastity, prove their virginity and to remain 'closed' for their husbands.[95] [98] [128] [129] [149] [152] This norm was so strong that some studies reported younger women wishing that they could undergo deinfibulation, experiencing unpleasant symptoms and being aware of services, but feeling unable to resist community pressure.[93] [129] [144] Some exceptions to this norm were reported, but mainly in the context of needing help for particularly difficult or painful symptoms.[95]

Care-seeking in relation to mental health issues associated with FGM/C was infrequently reported, and a few studies suggested that this may also be linked to cultural norms in which mental health is still often viewed as a stigmatised issue and, like FGM/C itself, is seen as hard to talk about.[93] [123] [128] [143] [150]

### Theme 2.2: influence of husbands and the family

Nineteen studies reported that access to care in terms of healthcare decision making, especially among Somali women, was strongly influenced by the views and advice of the wider family and peer group. In the maternity context, this had an impact on decision making around uptake of antenatal care and caesarean section.[91] [93] [95] [118] [132] [149] [150] The family also influenced decision making around FGM/C specifically. For example, as mentioned above, women considering premarital deinfibulation reported that they might avoid informing their direct family members as they were aware that they would otherwise come under strong pressure to change their mind due to fears over their future marital prospects.[96] [144] For married women with type 3 FGM, studies indicated that the views of husbands were of paramount importance and that women generally felt that they needed their husband's permission to undergo deinfibulation (outside of the context of delivery).[144]

### Theme 2.3: knowledge and information about FGM/C services

Women's knowledge of, and familiarity with, health services was variable. In some cases, women lacked familiarity with the host country health system, and this impeded their ability to access care in general. In other cases, women reported being quite familiar with maternity and primary care services, but lacked knowledge and information of FGM/C-specific specialist services (especially non-maternity-related services such as counselling) and where and when it may be appropriate to seek help.[93] [134] [143] One study from the UK suggested that this might be a particular issue for women living in low prevalence areas.[142]

### Theme 2.4: 'Hit & miss' care

When accessing healthcare, women reported that (1) the identification of FGM/C and (2) provision of appropriate treatment or referrals could be a 'hit and miss' process, depending on individual provider characteristics and practices, rather than being a result of standardised organisational systems and processes. For example, many studies cited situations where opportunities to identify FGM/C during a consultation were missed, which was attributed to providers lacking awareness of FGM (or related specialist services), and hence failing to ask the right questions, or to undertake an examination or to make timely referrals.[143] Maternity services were more likely to have knowledgeable providers and responsive reporting and follow-up systems, but even here women described situations where they felt their FGM/C had been mismanaged due to lack of awareness and lack of appropriate referrals.[96] [115] [141] [143] The most common issue was failure to have been asked about FGM/C so that it was not included in birth planning discussions and was identified only once labour had started.[92] This theme was reported in 38 studies across different countries and time periods.

### Analytical theme 3: cultural and bodily dissonance: striving for cultural and bodily integrity

This analytical theme refers to changes in and challenges to women's sense of cultural and bodily identity and integrity as a result of their experiences associated with FGM/C in the host country, and particularly through their encounters with the healthcare system (manifested, for example, during experiences of clinical examinations, childbirth, deinfibulation and reinfibulation). The theme describes women's responses to experiences of dissonance, whereby their decision making and actions can be understood as a desire to maintain a sense of cultural and bodily integrity. This theme highlights differences in, and changes to, cultural meanings and values related to FGM/C and how these differences and changes influence the way that women and health professionals define health issues, make decisions around FGM/C care and experience that care. When mutual interpersonal cultural understanding was not achieved, women perceived suboptimal care and distrusted the health system and their health providers. If women felt understood, they felt safe and reassured, resulting in a sense of bodily and emotional integrity.

### Theme 3.1: moving from normal to different

Some women described becoming aware of FGM/C as something that is 'different' only once they moved to another culture. At this point, some started to become aware of the suffering and symptoms that their FGM/C may have caused them. They started to question and to resist previously taken for granted aspects of their culture

and to feel uncomfortable with their own bodies and sexual identity.[100] However women also reported feeling shame and anger at being labelled as 'different' and 'mutilated' by the dominant discourse in the host country and by health professionals, and felt that their culture was misunderstood.[90 96 136 137] This experience of dissonance made some women feel reluctant to mention FGM/C to health providers or caused them to feel extremely uncomfortable during healthcare consultations, thereby limiting the opportunity to discuss the issues more openly and to achieve greater mutual understanding.[91 100 130] This theme was very common—reported in 40 studies.

### Theme 3.2: threat to the self: reliving FGM/C pain during clinical interventions

Many studies (n=28) reported women's experiences of gynaecological clinical interventions and childbirth as events which threatened their sense of bodily and cultural integrity. Women described great pain, suffering, fear and apprehension around clinical interventions—and related this to their previous healthcare experiences, and especially to reliving the original trauma they had experienced during their FGM/C. These emotions were exacerbated in healthcare encounters where women felt a loss of control or lack of respect. Negative experiences could lead to women avoiding attending the hospital until the last minute as a way of avoiding medical interventions.[155]

### Theme 3.3: being opened: complexities of deinfibulation for women with type 3 FGM

The experience of dissonance and quest for integrity was also seen in views and experiences around deinfibulation and was reported by 28 studies. As noted above, for cultural reasons, 'being opened' medically was seen as necessary primarily after marriage and primarily in the context of pregnancy and childbirth, rather than before.[121 122] There were no studies that explicitly or exclusively focused on the views or experiences of unmarried women, but some studies reported examples of unmarried women where the need to maintain cultural and bodily integrity outweighed the experience of physical pain or discomfort, even when women knew about the option to seek surgical deinfibulation.[93 128 129] There were few reported exceptions to this rule, although one study suggested that for some women, finding out about the option for surgical deinfibulation was empowering and deinfibulation was subsequently seen as a way of asserting control over their bodies and lives.[96]

Most studies reported that women preferred to be 'opened' during labour to avoid the pain and trauma of being cut twice.[91 95 98 120 129 144] However, some studies suggested that this view could change towards the medically preferred option of antepartum deinfibulation, if there were opportunities for women to have appropriate discussions, conducted in a sensitive way in the context of a trusting relationship with a healthcare provider.[130 134] Hence, given that FGM/C is often not discussed during healthcare consultations, women's current reported

preferences around deinfibulation timing may also reflect the fact that they may not be receiving the right information at the right time, and thus may not be in a good position to make informed decisions on this issue.[96 112 128 134]

### Theme 3.4: being changed: complexities around reinfibulation

A consistent finding across the studies within this theme (n=19) was that 'being opened' could involve significant emotional, physical, social and relational adjustments as women's bodies became changed. For some women, deinfibulation was firmly welcomed, although still required a period of 'getting used to' a different way of looking and feeling.[100 144] For other women, however (especially women from Sudan where it is a cultural norm in some regions to be closed again after delivery), there was ambivalence about these changes and some women requested to be 'closed again' to varying degrees.[94 96 100 118 128 129 144 146]

In most OECD countries, reinfibulation is illegal, and some studies reported women feeling very upset when their requests were denied by health professionals, experiencing this as a denial of their agency and integrity.[112 116] Consequently, women in one study reported seeking medical deinfibulation during holidays back home.[134]

### Analytical theme 4: disempowering care encounters

In many situations, the care received by women in relation to their FGM/C was experienced as fundamentally disempowering. Women described experiencing negative attitudes or behaviours from health professionals and poorly managed clinical interventions in which they felt they were not respected or listened to. Many studies reported examples of care encounters that left women feeling retraumatised, voiceless, without power to question interventions and vulnerable in the hands of unprepared system. Such experiences caused women to feel unsafe and disempowered. There were many examples where women felt that care had gone wrong or where experiences had been poor, and that these situations could have been avoided with better communication, cross-cultural understanding and sensitivity. This analytical theme is made of four descriptive themes that cut across all countries, time periods, population groups and care contexts.

### Theme 4.1: feeling exposed and humiliated

Across many studies (n=40), women reported that the reactions of healthcare professionals to their FGM/C had left them feeling ashamed, objectified, humiliated and exposed. This was primarily the case with healthcare professionals who were unfamiliar with FGM/C and who were encountering it for the first time. In these situations, women reported that healthcare professionals often reacted with extreme shock when they saw that a woman had been cut. Some tried to hide their shock and avoided mentioning anything at all, making the encounter feel uncomfortable. Others openly expressed horror or

disgust or reacted insensitively by asking intrusive or inappropriate questions. In many cases, other colleagues were called in to be consulted or to 'have a look' (sometimes without consent[119 130]), causing women to feel that their privacy had been violated and that they had been put on display. Such experiences left women feeling angry, distressed or as if they had been revioleted.[90 91] In some cases, these experiences left women wanting to avoid further contact with services and/or feeling unable to discuss their care any further as it was clear that the provider had limited knowledge about FGM/C.[148] Several studies made the point that negative experiences of a particular service or provider would then be discussed within peer and community networks, and hence could have an impact beyond the individual woman, potentially affecting the way others in the community might engage with care.[89 91 92 112 122 124 127]

### Theme 4.2: feeling judged and stereotyped

In many studies (n=36) women reported feeling judged and discriminated against within some of their healthcare encounters. They reported feeling that health providers made negative assumptions about them and provided suboptimal care based on racial, religious or other stereotypes and misconceptions about their culture in general, as well as about FGM/C. Such experiences caused distress, anger and avoidance of the health provider or service.[130]

### Theme 4.3: lack of choice, power and control

In this theme, reported by 32 studies, women described experiences where they felt they had lacked choice or control within the healthcare encounter, especially in the maternity setting and around key interventions such as caesarean sections or episiotomies. Lack of control was experienced in terms of feeling excluded from healthcare decision making, not being informed, not being listened to, feeling at the mercy of 'the system' and unable to express their needs. Women's attempts to assert control were sometimes construed by health professionals as being 'difficult'.[91 92 119 145 150] In labour, women described that FGM/C was often not discussed at all or that they were not consulted about interventions such as deinfibulation or caesarean section. This led to suspicion and fear, and was particularly reported in studies conducted with the Somali community.[90]

### Theme 4.4: feeling unsafe and vulnerable

Many studies (n=46) reported care encounters where women felt unsafe and vulnerable, primarily when they were treated by providers who they perceived to be inexperienced in dealing with FGM/C, and when they faced language barriers or were unable to express their own preferences. Such encounters often led to perceived poor-quality care or poor clinical outcomes.[139] The consequences ranged from providers failing to recognise, identify or discuss FGM/C,[89] to situations where women felt their care had been adversely affected as a direct result of poor provider skills (eg, having unnecessary caesarean

sections or extensive perineal tearing).[23 25 89 94 95 99 112 115 128 143 153] Women endured such situations feeling unsafe and highly vulnerable, often describing painful and traumatic experiences.[100] Such feelings of vulnerability were heightened for women who lacked social support.[90 95 130 140] In contrast, women reported feeling safe and highly reassured when they encountered providers who appeared to be knowledgeable or experienced.[89 92 94 96 100 101 135 139 143]

### Analytical theme 5: positive care encounters

As seen above, the synthesis has revealed many reports of poor care and difficult experiences in relation to FGM/C. However, many studies reported a mixture of experiences, including both good and poor care. High-quality care was conceptualised by women as care that is safe, respectful, culturally sensitive and compassionate. The essential process underpinning the experience of 'good' care was having a sense of trust—both in the individual provider and also in the 'system' as a whole. Trust was essential for, and developed from, positive care experiences. Positive experiences were associated with the development of a good relationship with health professionals who were perceived to be clinically knowledgeable and culturally sensitive, facilitating open communication and a sense of being in safe hands. Trust in the 'system' was linked to women's appreciation of the availability and accessibility of services (compared with their home countries), but particularly when services were perceived to be prepared to be responsive to community needs (eg, provision of appropriate interpretation services) and to manage issues relating to FGM/C (eg, by having specialist services or specialist providers).

### Theme 5.1: trusting and appreciating providers and the system

This theme, reported in 34 studies, showed that women appreciated the good medical services available in the host country. Women also expressed great appreciation for providers who made them feel safe and respected. Such providers were described as knowledgeable and experienced, who treated women with respect, who understood their individual needs and who involved them in their care. Women described these characteristics as facilitating trust and leading to open communication about FGM/C. For example, they were more likely to engage in conversations about their care (eg, birth planning), and in some cases were more likely to follow current medical guidelines, for example, on issues such as deinfibulation timing.[140] A few studies explicitly linked the ability to form good relationships to services that enabled continuity of care.[95 139]

### Theme 5.2: voicing healthcare needs and preferences

Women's recommendations for good-quality, safe FGM/C-related care and services they could trust were reported in 45 studies. They included (1) interpersonal provider characteristics and behaviours (such as providers being willing and able to talk about FGM/C, providers being skilled and knowledgeable around

FGM/C, and providers offering culturally sensitive and respectful care) and (2) service organisation issues (such as having specialist services for FGM/C, being given information and awareness about FGM/C-related services, and involving women and affected communities in FGM service development).

### Confidence in the review findings

Confidence in the review findings (as assessed by GRADE-CERQual) ranged from high to medium, and there was strong consistency in the findings across countries, population groups and clinical contexts (see table 6 for the final CERQual assessment for each theme and see online supplementary file 7 for the full details of the CERQual evaluation).

## DISCUSSION: ACHIEVING CULTURALLY SAFE CARE FOR WOMEN WITH FGM/C

The review shows that FGM/C—literally and symbolically—embodies complex processes of cultural change and cultural interaction that are occurring as a result of globalisation, migration and superdiversity. These processes manifest themselves very concretely in healthcare in terms of influencing how individuals seek care, how care is delivered and experienced by all involved, and how services are configured.[2 156 157]

This review has shown that, when obtaining care for FGM/C-related issues in a host country, women face common challenges that affect many groups of migrants, but that these are exacerbated.[46 84 158–161] Silence, secrecy, stigma and lack of familiarity with FGM/C within the system combine and act as obstacles to identifying FGM/C or to providing women with appropriate care. The care for FGM/C described in this review exemplifies the challenges of achieving 'cultural safety' in health services. Many of the experiences reported in the review describe a situation of physical but also cultural 'vulnerability' or 'risk' whereby women felt physically unsafe but also disrespected in terms of their cultural identity and bodily integrity and uninvolved in their care. Many of the themes in this review referred specifically to maternity care contexts, and there are striking similarities with themes identified in studies across the world that highlight disrespect and abuse in maternal healthcare.[35 36 162] However, as this review shows, the same challenges also appear to exist for women with FGM/C in primary care and other health settings.[163–165]

This review has also shown that, in some circumstances, women and providers can overcome these obstacles. Key to this are knowledge of services, community engagement, the availability of specialist services, continuity of care, shared decision making, addressing language barriers, and providing care that is person-centred, culturally sensitive and respectful.[44 159 166 167] Similar recommendations have also been made in other studies exploring the key elements of 'respectful care' in a variety of settings.[168 169]

Figure 2 outlines a model, derived from the review findings, of four key elements of 'culturally safe' care for women/girls with FGM/C, set within a wider context of superdiversity.

### Recommendations for culturally safe care

As per the model in figure 2, the review recommendations fall into four overlapping areas.

#### Overcoming barriers: information, awareness and community engagement

The review showed that there is a need for communities to be more aware of FGM/C-related services and of potential interventions. However, it also highlighted the challenges that women may have in discussing FGM/C within their family or community and with a health provider. This finding highlights a need for ongoing community involvement and engagement to raise awareness of services and provide support to women who may wish to access help (especially for unmarried or non-pregnant women).[31 32] The review suggests that service models which involve community advocates or community liaison workers may be particularly helpful in encouraging women to access care and to overcome communication difficulties with providers. Similar models have been found to be effective in other areas of healthcare.[170–174]

#### Breaking the silence: open communication and shared decision making

Given the communication challenges around FGM/C, the review findings strongly suggest that the onus of 'breaking the silence' may lie with the healthcare provider, with women stating that they expected and wanted health professionals to raise the issue. It may be appropriate for services to consider routinising questions around FGM/C in key clinical settings, as has been done for other 'sensitive' clinical issues such as domestic violence or HIV.[175–178] In addition, given the particular global concern for women's (and their babies') well-being related to timing of deinfibulation surgery (for women with type 3 FGM/C), the review findings strongly suggest that women appreciate, and would benefit from, more discussion, information and advice—at different stages in their life (before and after marriage) and both predeinfibulation and postdeinfibulation.

#### Provider competence and confidence

The review highlighted that providers may lack the confidence or cultural competence to raise the subject of FGM/C in an appropriate and respectful way, potentially missing opportunities to identify FGM/C in a timely manner but also missing opportunities to discuss prevention/safeguarding.[30] Communication challenges can be addressed through staff and interpreter training.[179] Likewise, the review showed that providers would benefit from additional training in associated clinical skills such as deinfibulation. The evidence base for best practice in health worker training around FGM/C is still weak[180]; however, as per WHO guidelines,[20] it is suggested that

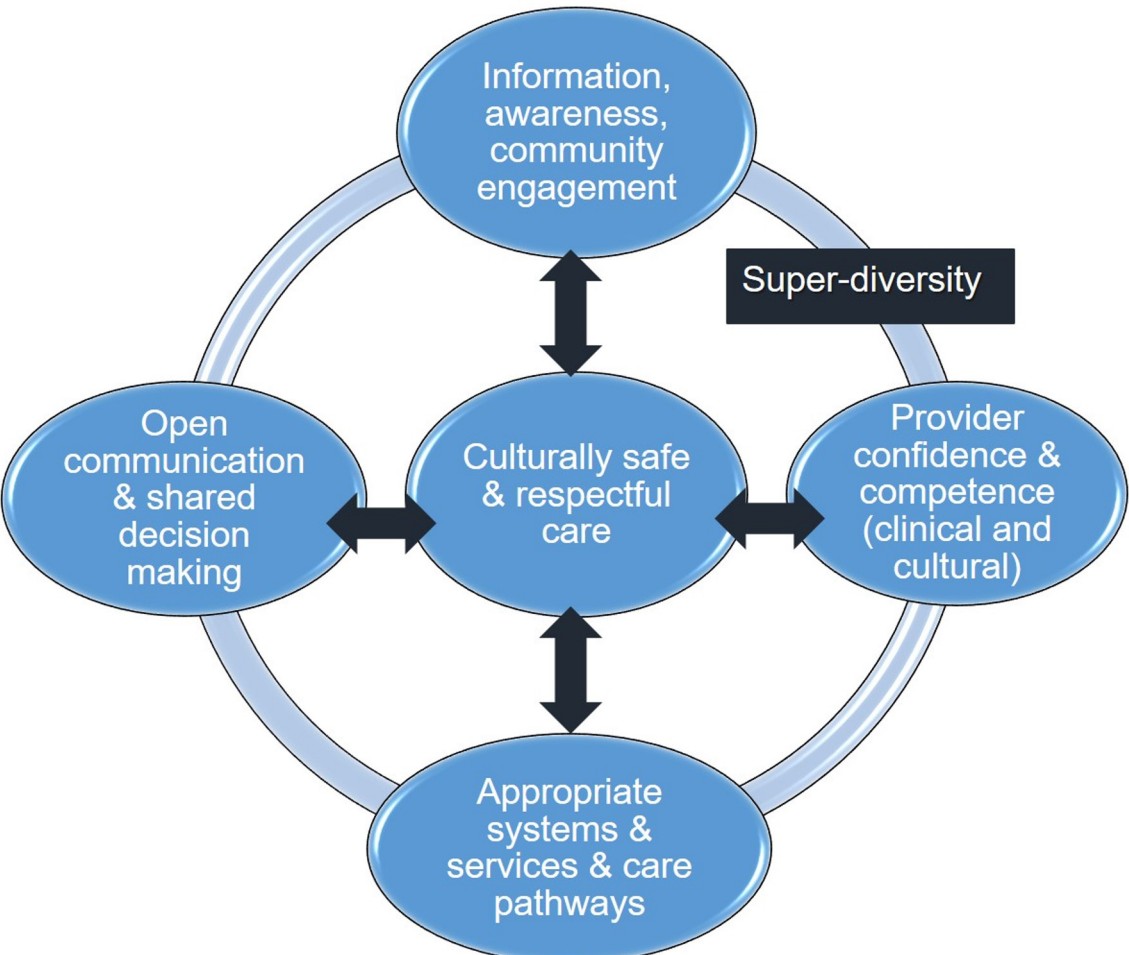

**Figure 2** Culturally safe care for female genital mutilation/cutting.

basic training around FGM/C should be embedded in preservice curricula for all health professionals, as well as being offered as continuous professional development for providers in more specialised roles.

### Getting the model of care right: appropriate systems, services and care pathways

The review suggests that women sometimes receive suboptimal care due to their FGM/C being missed or because of a lack of appropriate referral and reporting pathways. Some countries have made great strides to develop clear processes in this area (eg, UK), but in others service provision remains ad hoc.[21] As per WHO guidelines,[9 20] the review affirms the ongoing need to develop holistic care pathways.[13 26 181–184] The review findings suggest that women have better experiences in places where there are specialist services with confident and competent staff, and that women in lower prevalence or rural areas may be at a particular disadvantage. Hence, it is important for regions to develop service models that can address these inequalities. In addition, the review strongly suggests that respectful open discussions about FGM/C are more likely to happen in a context of a trusting relationship, and that such relationships are facilitated through service models that implement continuity of care approaches and shared decision making.[163–165 185]

The review has highlighted the ongoing need for the availability of mental health assessment and support where required. In particular, the review showed that for some women affected by FGM/C, childbirth and other clinical interventions could be traumatising, triggering flashbacks of their original cutting, creating profound anxiety and distress, and potentially leading to poor care experiences or avoidance of services.[186 187] The review was unable to illuminate in any further depth how such women might best be identified or treated, or which women might be most at risk. This is an important agenda for future research.[13 39 188–192] However, most maternity care services have well-established processes in place for dealing with highly anxious women or women who have experienced trauma. It would seem important for services to ensure that these interventions are also available for women affected by FGM/C.[35 193]

### Strengths and limitations

As mentioned above (under the Rationale section), previous reviews have primarily focused on care

 Evans C, *et al. BMJ Open* 2019;**9**:e027452. doi:10.1136/bmjopen-2018-027452

experiences within maternity contexts.[35][36][40][41] Our findings on care experiences resonate strongly with these existing reviews. However, we feel that our review has been able to expand on current knowledge in several ways. First, it adopted an exceptionally comprehensive and exhaustive search, and we are confident that the majority of relevant papers were identified. The inclusion of grey literature as well as studies in other languages significantly broadened the evidence base (eg, the existing reviews on maternity contexts included between 4 and 16 papers, whereas this review included 57).[35][36] It is our contention that this wider evidence base, as well as the broader review focus on all aspects of the life course, has enabled a more nuanced understanding to emerge around FGM/C-related health experiences. In particular, our review has identified new findings around (1) factors that influence care-seeking and access to care (as well as the care experience itself), (2) decision making around deinfibulation surgery, (3) community perspectives on positive care experiences and suggestions for service improvement, and (4) being able to demonstrate the importance of primary care as well as maternity contexts for discussing and identifying FGM/C. The theme matrix presented in online supplementary file 4 provides a visual means for exploring in more detail how the wider evidence base has contributed to particular review findings, and thus how it has enabled new or more nuanced themes to be identified.

Another strength of the review is its exclusive focus on OECD countries, which ensures that its findings are highly relevant to health service development in most destination countries.

The main limitation of the review is that the findings inevitably reflect the methodological limitations of its included studies. Many papers displayed a homogenising tendency in terms of the affected communities as well as FGM/C types. For example, in papers that had mixed community samples, there was little attempt to explore cultural differences in relation to FGM/C. Likewise, it was not always clear which type of FGM/C was being studied. In research with Somali or Sudanese communities, the assumption can reasonably be made that their focus was FGM/C type 3. However, in studies with mixed samples, there was very little differentiation on the basis of FGM/C type. As a result, there is limited specific knowledge on the needs or experiences of women with other FGM/C types.

### Evidence gaps and future research

The review has identified five key gaps in the evidence base around FGM/C, all of which are priority areas for future research. First, there were very few studies that focused specifically on mental health needs. Second, there is a lack of studies on girls' or unmarried/non-pregnant women's experiences around FGM. Important areas for future investigation among this group are care-seeking and decision making (especially

around deinfibulation timing). Third, there is a lack of knowledge on potential health needs or experiences of older women; hence, this is an area that may require further investigation. Fourth, there were no indepth evaluations of interventions or services, which means that development of models of 'good care' (eg, refs 140 194) still needs to be inferred rather than being based on sound evidence. Further mapping and evaluation of models of care would be beneficial to understand better their differential impact on accessibility, outcomes, cost and patient satisfaction. Finally, there was no research that specifically explored women's/girls' experiences of healthcare professional's involvement in the implementation of legal safeguarding or prevention interventions, which in some countries (such as the UK) are inserting new complexities into the patient–professional relationship and reportedly undermining trust.[32][33][195–197]

### CONCLUSION

This review has identified key challenges but also opportunities for the development of culturally safe and accessible services to improve care for women affected by FGM/C. Future research should involve communities to evaluate existing models of care in order to inform best practice.

**Acknowledgements** We would like to thank all those who have contributed time and valuable insights to this project. We would particularly like to thank all those who gave up their day to attend the national stakeholder consultation event. Special thanks go to Carol McCormick (FGM/C specialist midwife) for her support and input at the start of this project. We gratefully acknowledge the invaluable contributions of our project advisory group: Kinsi Clarke, Nottingham and Nottinghamshire Refugee Forum, cofounder of Nottingham FGM survivor's group; Professor Jim Thornton, consultant obstetrician, Nottingham University Hospital Trust and University of Nottingham; Amanda Wickham (FGM/C specialist midwife), Nottingham University Hospital Trust; Helen Denness, consultant in public health, Nottingham City Council; and Grace Brough, insight specialist (public health), Nottingham City Council, Chair of Nottingham and Nottinghamshire FGM/C Steering Group.

**Contributors** CE and VN conceived the study. CE obtained funding, oversaw all aspects of the project and contributed to all stages. CE drafted this paper. RT contributed to all stages of the review, including data extraction, coding, quality appraisal and CERQual components. JE designed and executed all the searches. JM assisted with quality assessment and contributed to the evolving synthesis. GMAH assisted with quality assessment and contributed to the evolving synthesis. JA provided clinical expertise and contributed to the evolving synthesis and formulation of recommendations/conclusions. VN provided a community perspective and contributed to the evolving synthesis and formulation of recommendations/conclusions.

**Funding** This project was funded by the NIHR HS&DR Programme (no 15/137/04). The views expressed in this report are those of the authors and not necessarily those of the NHS, the NIHR or the Department of Health and Social Care.

**Competing interests** None declared.

**Patient consent for publication** Not required.

**Provenance and peer review** Not commissioned; externally peer reviewed.

**Data sharing statement** No additional data are available.

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
