## [Reviewer comments · BMJ Open]

ARTICLE DETAILS

TITLE (PROVISIONAL)	Seeking Culturally Safe Care: A Qualitative Systematic Review of the Healthcare Experiences of Women and Girls who have Undergone Female Genital Mutilation/Cutting
AUTHORS	Evans, Catrin; Tweheyo, Ritah; McGarry, Julie; Eldridge, Jeanette; Albert, Juliet; Nkoyo, Valentine; Higginbottom, Gina

VERSION 1 - REVIEW

REVIEWER	Angela Dawson The Australian Centre for Public and Population Health Research, Faculty of Health, University of Technology Sydney
REVIEW RETURNED	07-Jan-2019

GENERAL COMMENTS	This comprehensive review contributes a life course approach in countries of migration and draws upon literature in languages other than English. I think it is useful addition to the literature. It is pleasing to see the use of the CERQual tool that adds another layer of methodological rigor to this review. I have a few comments. “This review is informed by the theoretical construct of ‘cultural safety’” How did this inform the data analysis and synthesis? While the authors talk about a “code book” it is unclear what these codes were and how they formed categories and themes. Can the authors please elaborate with some examples? While the authors set their review apart from others they make no direct comparison with their findings and others in terms of the identified themes. What specifically have the 4 non English studies added that sets this review apart from others? “Figure2: Culturally safe care for FGM/C” The link between this model and the recommendations is not clear and needs further clarification. “Evidence gaps and futures research” These are important gaps identified by this review. What about the health care needs of older women with FGM?
---

REVIEWER	Ingvil Krarup Sørbye Oslo University Hospital Norway
REVIEW RETURNED	22-Jan-2019

GENERAL COMMENTS	General comments: The paper is a qualitative systematic review of FGM-related healthcare experiences of women/girls living in OECD-countries.
---

	The study has a clear purpose, is well-designed and the methodology used is robust. The protocol has previously been published in BMJ Open (reference 34). The conclusions and recommendations are based on findings. These are clear and well-formulated and relevant to clinical service providers. The scarcity of studies among children and older women makes conclusions/recommendations difficult for these groups. Thus the objective of assessing women FGM-related care across her life-span is only partly accomplished. Minor revisions: 1. Page 36: Recommendations- Getting care pathways right Clear and holistic care pathways are recommended at system-level. However, in contexts where this has been established, issues as to poor attitudes and level of knowledge among individual health care staff can remain, making systems ineffective. Based on your results, do you have a recommendation as to how this best should be addressed; e.g. inclusion in health staff curricula, post-graduate training or other? 2. Reference 33: The registration number differs from the one registered in Prospero (accessed 20.01.19).
--	--

VERSION 1 – AUTHOR RESPONSE

Response to Reviewers' Comments

Reviewer 1 Comments	Response to Reviewer 1
“This review is informed by the theoretical construct of ‘cultural safety’ How did this inform the data analysis and synthesis?	Thank you for this excellent question. We have added in a paragraph under the heading ‘Theoretical perspective’ (p.6-7) explaining our position and approach in much more detail.
While the authors talk about a “code book” it is unclear what these codes were and how they formed categories and themes. Can the authors please elaborate with some examples?	We have added in some more detail on the coding and analysis process into the main paper (on p.12). We felt that an even fuller description (using examples) would add too many words to the main paper and may not be of interest to all potential readers. To get around this, we have created a new Supplementary File (no.3) in which the whole process is described in considerable detail and illuminated with examples.
While the authors set their review apart from others they make no direct comparison with their findings and others in terms of the identified themes. What specifically have the 4 non English studies added that sets this review apart from others?	We have re-worked the ‘Strengths and Limitations’ section to address this question more specifically (see also the ‘Rationale’ section). However, we would respectfully argue (as we have done in the paper) that it is not just the inclusion of non-English language studies,

	but also grey literature and a wider overall review focus that has enabled new understandings and new themes to emerge. These have now been specified more clearly in the paper. Some of the grey literature or non-English studies simply confirmed existing findings, however others produced new findings. A detailed examination of the Theme Matrix in Supplementary File no. 4 will hopefully illustrate the contribution of respective papers to different review findings more clearly.
"Figure2: Culturally safe care for FGM/C" The link between this model and the recommendations is not clear and needs further clarification.	Thank you for highlighting this issue. We have restructured the recommendations to show how they link directly to the domains within the model.
"Evidence gaps and futures research" These are important gaps identified by this review. What about the health care needs of older women with FGM?	This omission has now been included in the section on 'Evidence gaps and future research'.
Reviewer 2 Comments	Response to Reviewer 2
Page 36: Recommendations- Getting care pathways right. Clear and holistic care pathways are recommended at system-level. However, in contexts where this has been established, issues as to poor attitudes and level of knowledge among individual health care staff can remain, making systems ineffective. Based on your results, do you have a recommendation as to how this best should be addressed; e.g. inclusion in health staff curricula, post-graduate training or other?	Thank you for this suggestion. We have added in a specific paragraph around training for providers in the Recommendations section (under a new heading 'Provider competence and confidence').
Reference 33: The registration number differs from the one registered in Prospero (accessed 20.01.19).	Thank you for pointing out this error. We have corrected it and inserted the correct PROSPERO record number (now Reference number 69).

VERSION 2 – REVIEW

REVIEWER	Ingvil Krarup Sørbye Oslo University Hospital Norway
REVIEW RETURNED	22-Mar-2019
GENERAL COMMENTS	The Authors have adequately addressed my concerns. The revised manuscript is satisfactory.